# Minimally Invasive Plate Osteosynthesis (MIPO) of Comminuted Radial Fractures Using a Locking Plate Contoured on a 3D-Printed Model of the Feline Antebrachium: A Cadaveric Study

**DOI:** 10.3390/ani14091381

**Published:** 2024-05-04

**Authors:** Piotr Trębacz, Jan Frymus, Mateusz Pawlik, Anna Barteczko, Aleksandra Kurkowska, Michał Czopowicz

**Affiliations:** 1Department of Surgery and Anaesthesiology of Small Animals, Institute of Veterinary Medicine, Warsaw University of Life Sciences-SGGW, Nowoursynowska 159 C, 02-776 Warsaw, Poland; 2CABIOMEDE Ltd., Karola Olszewskiego 21, 25-663 Kielce, Poland; mateusz.pawlik@cabiomede.com (M.P.); anna.barteczko@cabiomede.com (A.B.); aleksandra.kurkowska@cabiomede.com (A.K.); 3Department of Biomaterials and Medical Devices Engineering, Faculty of Biomedical Engineering, Silesian University of Technology, Roosevelta 40, 41-800 Zabrze, Poland; 4Division of Veterinary Epidemiology and Economics, Institute of Veterinary Medicine, Warsaw University of Life Sciences-SGGW, Nowoursynowska 159 C, 02-776 Warsaw, Poland; michal_czopowicz@sggw.edu.pl

**Keywords:** additive manufacturing, fracture reduction under the plate, cat

## Abstract

**Simple Summary:**

Three-dimensional printing turns a digital model in a computer file into a physical interpretation of the object. This technology is promising in both human and veterinary medicine, enabling the production of anatomical models, customized surgical instruments, implants, and prostheses. Minimally invasive plate osteosynthesis (MIPO) techniques have been developed to accelerate fracture healing by preserving soft tissue as well as the environment at the fracture site. Achieving acceptable alignment with MIPO can be challenging, because fractures are typically reduced indirectly and implants are inserted submuscularly, starting at one end of the broken bone. In our study, we presented results of comminuted feline radial fracture repair using MIPO with locking bone plates as a reduction tool. The implants were pre-contoured on 3D-printed bone models of the antebrachium of an adult male and a female domestic shorthair cat. A pre-contoured bone plate is a valuable tool for fracture reduction in MIPO.

**Abstract:**

(1) Background: Due to the unique structural and functional characteristics of the forelimb in cats, fractures of the radius and ulna are best repaired using internal fixation and stabilization in accordance with AO principles. This study presents the results of reduction of 42 cadaveric comminuted feline radial fractures reduced by minimally invasive plate osteosynthesis (MIPO). (2) Methods: Radius fractures were created on 21 pairs of forelimbs with intact bones. MIPO was then performed using two locking bone plates pre-contoured on 3D-printed bone models of the antebrachium of a male and a female cat. Pre- and postoperative radiographs were taken, and radius length and anatomical lateral distal radial angle (aLDRA) were measured. (3) Results: All fractures were classified as complex diaphyseal fractures of the radius. The radial bone length did not change significantly after surgery (F_1,18_ = 0.01, *p* = 0.933). However, the aLDRA was modified after surgery (F_1,18_ = 7.51, *p* = 0.013), but this change was only observed in females, in whom the aLDRA was significantly reduced (*p* = 0.035) compared to the value determined by the shape of the plate. In males, the pre- and postoperative aLDRA values were similar (*p* = 0.824). In 40 cases, alignment, adjacency of bone fragments, and apparatus were judged to be satisfactory. In two cases, the plate was fixed to the proximal radius and distal ulna due to misidentification of the distal radius. In both cases, revision surgery and correct fixation of the radius gave proper alignment, adjacency, and apparatus. (4) Conclusion: A pre-contoured plate on a 3D-printed model of the male and female domestic cat antebrachium was suitable for the reduction and stabilization of comminuted radial fractures in a cohort of domestic cat cadavers without the need to print individual antebrachial bone models for each patient.

## 1. Introduction

The incidence of antebrachial fractures in cats is relatively low. They occur in 2–14% of feline patients [1,2]. Similar results were found by Cardoso et al. [3], who evaluated 179 long bone fractures in 141 cats and found that radius and ulna fractures accounted for 11% of all fractures analyzed. In this study, 16% of forearm fractures were comminuted. Antebrachial fractures in cats are rare, and they can be difficult to treat. This is particularly true of comminuted fractures. Cats have unique structural and functional characteristics of the forelimb that can influence clinical outcomes after repair of antebrachial fractures. Their interosseous membrane connecting the radius and ulna is more pliable and extensive than that of a dog, contributing in part to approximately twice the range of supination and pronation. Similarly to humans, cats need to maintain their antebrachial range of motion to perform daily functions. This means that fractures of the radius and ulna in cats should be treated surgically with appropriate fixation of both bones to reduce the risk of malunion or nonunion [4].

Minimally invasive plate osteosynthesis (MIPO) is a surgical procedure that respects the biology of the bone and soft tissue [5,6,7]. Fractures are usually reduced indirectly and implants are inserted epiperiosteally from one end of the bone, leaving the local fracture site undisturbed. Achieving acceptable alignment using MIPO can be challenging [8,9,10,11]; therefore, the basic principle while performing this surgery is to have a properly pre-contoured plate. Several indirect fracture reduction techniques have been described to facilitate the application of MIPO [10,12,13,14,15]. The implant can be contoured based on the surgeon’s experience, radiographs of the contralateral intact bone [13], or using 3D-printed bone models. Use of the latter has resulted in accurate surgical reconstruction of even complex fractures and good functional patient recovery in both humans and small animals [10,11].

The surgical approach to the feline antebrachium for MIPO differs from that used in dogs because the feline radius has a more pronounced lateral torsion. Therefore, a lateral approach is needed in cats for more direct exposure. In cats, the minimally invasive approach to plating the proximal radius is accomplished by creating an approach between the common digital extensor and the lateral digital extensor muscles. A supinator muscle may be elevated for a wider approach. The distal approach to the radius is made between the extensor carpi radialis tendon and the common digital extensor tendon [16,17].

Studies investigating the geometric properties of the feline skeleton are limited [18,19,20,21,22], but the differences in bone morphology between cat breeds are relatively small. Therefore, it may be easier to standardize implants for fracture fixation in cats than in dogs.

In domestic cats, the radius and ulna are similar in length and shape among individuals, with the exception that the radius is longer in males [18,21]. Therefore, we believe that the bone plate used as a reduction tool does not need to be pre-contoured for each individual. The aim of our study was to present the results of the reduction in 42 comminuted feline radial fractures in the MIPO using locking bone plates as a reduction tool. The plates were pre-contoured on 3D-printed bone models of the antebrachium of a random adult male and female DSH (Domestic shorthair).

## 2. Materials and Methods

### 2.1. Development of 3D-Printed Bone Models

Segmentation and 3D modeling of the forearms were performed using Digital Imaging and Communications in Medicine (DICOM) files of the bone volume imported into the computer software ITK-SNAP 4.0.2. (itksnap.org) (accessed on 29 April 2024). Computed tomography (CT) scans were performed using a helical scan technique with a slice thickness of 0.625 mm, following a bone-specific protocol to optimize visualization of skeletal structures. Three-dimensional representations were generated using the Hounsfield unit density range for bone (+226 to +2788) based on masks marked on 2D slices of the scan. One adult male and one adult female DSH underwent a whole-body CT scan after a road traffic accident. The antebrachial bones were reconstructed using a bone algorithm for evaluation and measurement. Materialise Magics 27 (Materialise NV, Belgium) was used to create the 3D printing machine code for the forearm models. The printing process used digital light processing (DLP) technology with resin and custom modification of the 3D printer based on Sonic Mini 8K (Phrozen Tech Co. Ltd., Hsinchu City, Taiwan). The right and left antebrachia were printed with radial lengths of 103 mm for the male and 92 mm for the female cat. The anatomical lateral distal radial angle (aLDRA) was measured on the models as described by De Lima Dantas et al. [22]. The aLDRA was 91.6° for the male radius and 92.4° for the female radius (Figure 1 and Figure 2). After printing, the models underwent a series of post-processing steps. These included thorough cleaning with isopropyl alcohol to remove any residual uncured resin and the precise removal of mechanical supports, which were critical in maintaining the integrity of the models during the printing process (Figure 3 and Figure 4).

### 2.2. Contouring of the Bone Plates

Two 2.4/2.7 locking plates (Medgal vet, Księżyno, Poland), 82 mm long (10 holes) for females, and 90 mm (11 holes) for males, were contoured to the 3D-printed models of the feline antebrachium with radial lengths of 92 mm and 103 mm, respectively. The plates covered the bones from the proximal to the distal metaphysis. Left and right plates did not have to be contoured separately, as they did not exhibit much deviation in curvature (Figure 5).

### 2.3. Preparation of Surgical Specimens

Surgery was performed on 21 pairs of forelimbs amputated at the scapula. Limbs were obtained postmortem from 21 adult cats, 12 DSH and 9 DLH, 11 males and 10 females, of unknown desexing status, aged from 3 to 16 years (9.5 ± 3.2 years), and weighing from 3 to 5 kg (4.0 ± 0.5 kg). Males were significantly heavier than females (4.3 ± 0.4 kg vs. 3.6 ± 0.4 kg; *p* < 0.001), but body weight did not differ significantly between the breeds (*p* = 0.463). The cats had no obvious skeletal pathology, and died or were euthanized for reasons unrelated to this study. Ethics approval was not required for this study according to Polish legislation (Act of the Polish Parliament of 15 January 2015 on the Protection of Animals Used for Scientific or Educational Purposes, Journal of Laws 2015, item 266). Comminuted fractures of the radius and ulna were created using a bone chisel and mallet.

### 2.4. Radiographic Assessment

Standard calibrated orthogonal radiographs of the amputated limbs were assessed before and after radial fracture reduction. For each limb, the length of the radius on the preoperative and postoperative lateral radiographs and the fracture configuration on the lateral and craniocaudal radiographs were assessed. In addition, anatomical lateral distal radial angle (aLDRA) was measured before and after osteosynthesis in the same way as described above for the bone models. Alignment (axial and rotational alignment of the joints above and below the fracture), adjacency (adequate attachment of the main fracture fragments to the comminution zone), and apparatus (location, placement, and the size of the implants) were also assessed on postoperative radiographs.

### 2.5. Surgical Procedure

The distal approach to the radius was first performed as described by Schmierer and Pozzi [16]. After creating the epiperiosteal tunnel with Metzenbaum scissors, the proximal approach was performed with additional elevation of the supinator muscle. After insertion of the plate, the first ∅ 2.4 mm cortical screw was tightened at the proximal end of the plate at the level of the radial neck. The bone fragment was then reduced under the plate using bone-holding forceps and the ∅ 2.4 mm locking screw was inserted into the second hole (Figure 6). The initially inserted cortical screw was then removed and the ∅ 2.4 mm locking screw was tightened in the same hole. After manual traction of the antebrachium and manipulation of the paw, the radial fracture was reduced and the distal end of the plate was screwed to the distal fragment of the radius in the same manner as described above (Figure 7).

### 2.6. Statistical Analysis

The distribution of numerical variables (age, body weight, and radial bone dimensions i.e., radial bone length and aLDRA) was assessed using normal probability Q-Q plots, and the normality assumption was verified with the Shapiro–Wilk W test. As the assumption was satisfied, numerical variables are presented as the arithmetic mean, standard deviation (± SD), and range, and analyzed with parametric statistical tests. The unpaired groups (sexes, breeds) were compared using the two-sample t-test. A generalized linear model (GLM) was applied to investigate the differences in dimensions of radial bones between breeds (DSH and DLH) and sexes (males and females) separately, controlled for sides (left and right introduced as paired groups). Significant differences were presented as the mean difference (± SD) with the 95% confidence interval (CI 95%). Coefficient of variation (CV) with CI 95% [23] was calculated to examine the variability in radial bone dimensions in the study population. The GLM was also used to analyze the change in radial bone dimensions (averaged for the left and right forelimb) after the surgery, and control groups for both sex and breed. A post hoc analysis in the GLMs was performed using Tukey’s test for unequal groups. Categorical variables are presented as counts in groups and compared between groups using Fisher’s exact test. All statistical tests were two-sided and the significance level (α) was set at 0.05. The minimum size of each group analyzed in the study was 9 pairs of forelimbs. This group size ensured ≥90% statistical power of comparisons of the radial bone length and the aLDRA between unpaired groups (i.e., males vs. females, DSH vs. DLH) as well as between paired groups (left vs. right forelimb, measurements before vs. after surgery). The minimum difference that could be detected at this group size was a difference in the mean radial bone length between groups equal to 5 mm (assuming SD in each compared group equal to 3 mm) and a difference in the mean aLDRA between groups equal to 5° (assuming SD in each compared group equal to 3°). The statistical analysis was performed using TIBCO Statistica 13.3 (TIBCO Software Inc., Palo Alto, CA, USA).

## 3. Results

The radial bone was significantly longer in males than in females (F_1,18_ = 180.10, *p* < 0.001) with a typical difference of 8–12 mm. On the contrary, the aLDRA was similar between sexes in the post hoc analysis despite an initial significant result of the omnibus GLM (F_1,18_ = 5.97, *p* = 0.025). Variability in radial bone dimensions within a certain sex was very low (Table 1).

There was no significant difference between the two breeds in radial bone length (F_1,18_ = 1.84, *p* = 0.192) or in the aLDRA (F_1,18_ = 0.25, *p* = 0.626).

### Surgical Procedure

All fractures were classified as complex diaphyseal fractures of radius and ulna: 22-C2 or 22-C3 in the AO VET fracture classification system [24]. In 40 cases, the axial and rotational alignment of the joints above and below the fracture was appropriate and attachment of the main fracture fragments to the comminution zone was adequate. Location, placement, and the size of the implants were all assessed as correct on postoperative radiographs. In two cases (two right limbs), the plate was fixed by mistake to the proximal radius and distal ulna (Figure 8 because the distal approach was made between the lateral digital extensor tendon and the carpi ulnaris tendon, rather than between the extensor carpi radialis tendon and the common digital extensor tendon. After revision surgery and correct fixation of the radius, the alignment, adjacency, and apparatus were all satisfactory.

The radial bone length did not change significantly after the surgery (F_1,18_ = 0.01, *p* = 0.933) (Figure 8). However, the aLDRA changed significantly after the surgery (F_1,18_ = 7.51, *p* = 0.013), but the change was observed only in females in which the aLDRA was significantly reduced (*p* = 0.035) compared to the value determined by the shape of the plate. In males, the pre- and postoperative values of aLDRA were similar (*p* = 0.824) (Figure 9).

## 4. Discussion

Musculoskeletal injuries are by far the most common cause of orthopedic problems in feline patients; therefore, the results presented in this study can prove to be of significant value to small-animal surgeons. We found that the two 82 mm and 90 mm 2.4/2.7 locking plates pre-contoured on 3D-printed bone models were very useful for the reduction and fixation of 42 complex radial fractures in the cohort of 21 domestic cat cadavers.

The bone plate can be used as a fracture reduction aid. During fracture repair, straight plates can be useful in reducing any relatively straight bony parts: however, bones with more complex shapes should be reduced with a pre-contoured plate. Currently, plate contouring is usually performed during surgery. However, it is impossible to clearly visualize the entire 3D bone structure, and it is difficult to repeat the contouring and positioning of the plate on the bone surface to confirm that it fits well, due to anatomical obstacles and reduction tools. As a result, plate contouring is generally considered to be an imprecise and time-consuming process. Human surgeons have attempted to solve this problem, and they often contour plates preoperatively on commercially purchased bone models when operating on complex anatomical structures such as the pelvis. However, as the commercially available bone models are made based on average human bone size, they cannot accurately reproduce each patient’s anatomy [11]. This is even more true for dogs. However, in most domestic cat breeds, the body size is very similar, and indeed our study confirmed that in our 21 randomly selected feline cadavers, the pre-contoured plates for both “male” and “female” plates were suitable for all queens and males.

MIPO was first described in humans in 1997 for distal femoral fractures [25]. Indirect reduction and percutaneous fixation of the plate in MIPO protects the bone blood supply and fracture hematoma and therefore appears to be beneficial for bone healing [7,14,26,27]. In the MIPO technique, the pre-contoured plate can be used as a tool to achieve axial realignment and correct rotations, angulations, and displacements. The advantages of MIPO vary depending on the fracture site. Bones with limited regional soft tissue, such as the radius and tibia, may benefit from preserving the soft tissue sleeve [7]. In addition to the ability to use a pre-contoured plate as a fracture reduction tool, optimal plate contouring minimizes soft tissue irritation and dead space under the plate. Plate-to-bone distance can also have an effect on the stiffness of the construct. Ahmad et al. [28] reported plastic deformation and failure at lower loads in 4.5-mm locking compression plate (LCP) constructs with a 5 mm plate-to-bone distance compared with a 2 mm plate bone distance. In our study, cortical screws and bone reduction forceps were used to reduce the fracture after the insertion of the well-contoured plate. As the screws were tightened, the plate came into contact with the displaced fragments, pushing and holding them in a reduced position. Bone-holding forceps were used to further stabilize the fragment during screw tightening. Finally, the plate was fixed to the bone with two locking screws proximally and two distally. This technique was found to be effective in all cases.

Bridging plate osteosynthesis is a well-accepted technique for treating comminuted fractures of long bones in small animals [7,13,14,27,29]. When locking plates are used for comminuted fractures as bridging plates in non-osteoporotic human bone, it is recommended that at least three to four monocortical screws or two bicortical screws per fragment [30,31] be used. This recommendation may be impractical in veterinary practice, as load transfer during locomotion and behavior of the animal patient is different from human patients. Cats are relatively strong and active animals, and two bicortical locking screws through the main bone fragment may not be suitable for load transfer in the postoperative period. In this study, two bicortical locking screws per main radial fragment were used to simplify the surgical procedure. We believe that more than two screws per main fragment would not affect the assessment of the suitability of the pre-contoured plate for fracture reduction. Palierne et al. [31] compared plates fixed with two and three locking screws per fragment and found that the bending stiffness of the two-screw constructs was lower than that of the three-screw constructs. Therefore, it is better to always try to use three bicortical locking screws per fragment. The accuracy of reduction in each radius was assessed by analysis of postoperative alignment, adjacency of bone fragments, and apparatus (plate and screws). We also measured the total length of the radius and aLDRA as an indicator of radius varus or valgus on pre- and postoperative radiographs. We could not measure the anatomical caudal proximal angle (aCdPRA), anatomical distal radial angle (aCdDRA), or sagittal procurvatum (SP), as suggested by De Lima Dantas et al. [22]. This was due to the presence of the plate, which made it difficult to measure such angles. In our cohort of cats, radial bone length did not change significantly after surgery. However, the aLDRA did change after surgery, but only in females. In female cadavers, the aLDRA was greater than the aLDRA of the 3D-printed bone model of the female radius (93.4° vs. 92.4°) and was reduced postoperatively to the value determined by the shape of the plate pre-contoured on such a model. The clinical significance of this small reduction in aLDRA has not been studied. In fact, it is rare in clinical practice to measure the bone angle before and after surgery in patients treated with MIPO. Fracture alignment can be assessed by two main techniques: intraoperative imaging, such as fluoroscopy, or clinical assessment [13]. The most common practice is to assess the restoration of bone length and alignment of the adjacent joints compared to the contralateral limb after surgery without intraoperative imaging. As torsion of bone fragments is one of the main difficulties in MIPO, more advanced imaging techniques such as computed tomography may be used for more appropriate postoperative assessment of bone alignment. In our study, we did not use computed tomography for postoperative assessment, and this could be considered a potential limitation of this study. There were only two intraoperative complications in our study. In both cases, the plate was fixed to the proximal radius and distal ulna due to misidentification of the distal radius and consequently misidentification of the surgical approach. After revision surgery and correct fixation of the radius, alignment, adjacency, and apparatus were accurate. The distal approach to the feline radius is close to the distal ulna, so correct anatomical identification of both bones is essential in order to avoid mistakes. The use of fluoroscopy to assess alignment and implant placement intraoperatively can reduce the risk of such complications, but fluoroscopy is not widely used in veterinary surgery and exposes the surgical team to ionizing radiation [13,29].

Modern diagnostic imaging, such as computed tomography, allows precise fracture analysis and preoperative planning of the osteosynthesis, thus expanding the indications for MIPO. Technological advances have significantly improved medical treatment practices, and 3D-printing techniques have been applied to the field of orthopedics. In simple terms, additive manufacturing (3D printing) is the process of taking a digital file and turning it into an object in the real world. Recently, several studies have shown that 3D printing can be used for efficient preoperative plate contouring based on the surface of the individualized bone model [10,11]. The feline radius is generally a straight bone with gentle curvatures. In our study, we found that a contoured plate of a 3D-printed model of the antebrachium of an adult male and female domestic cat was suitable for minimally invasive reduction and stabilization of comminuted radial fractures in domestic cats. This was achieved without the need to print individual antebrachial bone models for each patient. A control group with pre-contoured plates was not created in this study. Comparing two groups of animals, one treated with plates pre-contoured on 3D-printed models and one treated with plates pre-contoured for each individual, may provide valuable results and may be the subject of further studies. The use of intact forearms constitutes another limiting factor in this study. The fractures were created using a bone chisel and mallet. As a result, we obtained fractures with a configuration similar to naturally occurring fractures in cats, because many radius fractures are accompanied by ulna fractures [4]. Wallace et al. [32] reported that of a total of 48 antebrachial fractures, 28 were combined diaphyseal radial and ulnar fractures (58% of all antebrachial fractures presented). In all our cases, the fractures created were classified as 22-C2 or 22-C3 in the AO VET fracture classification system, which met our preoperative expectations. Such complex fractures are most commonly seen in cats that have suffered serious trauma, such as falls or road traffic accidents. In such patients, the MIPO technique is always favorable for the reduction and stabilization of these fractures, as MIPO is less time-consuming and less traumatic than open reduction and internal fixation (ORIF) [13,14,15].

The absence of post-traumatic changes associated with the fracture, such as swelling, hematoma, and skeletal muscle contracture, may also be considered a limitation, as reduction in fractures in cadavers without these changes was easier. Treatment of diaphyseal fractures with MIPO does not require anatomical reduction of bone fragments. The aim is functional reduction. It is important to restore bone length and alignment in the frontal, sagittal, and axial planes. In our study, initial reduction in fractures required only manual traction of the antebrachium and manipulation of the paw. Final reduction was performed after insertion and fixation of a pre-contoured plate. In clinical cases, especially in patients with severe skeletal muscle contractures, more forceful reduction methods should be used prior to final reduction of the fracture under the plate, such as the hanging-limb technique, intramedullary pinning, skeletal traction table [15], or external fixator [33]. For radial and ulnar fractures, it is possible to reduce the fractures indirectly after intramedullary pinning of the ulna. In this study, ulnae were not reduced or stabilized in order to test the usefulness of a contoured radial plate as a reduction tool; however, ulna fractures in cats should be stabilized. In most cases, the ulnar plates may be a better choice than intramedullary pins. Stabilization of the radius only in cats with mid-antebrachial fractures using type II external skeletal fixators has been reported as a risk factor for the development of non-union [2]. Also, the use of external coaptation as the primary method of fracture stabilization in feline antebrachial fractures was associated with a high percentage of development of non-union in one report [34]. Among domestic animals, only cats are able to actively perform significant supination movements. Dogs can only rotate the forearm passively. In addition, the increased range of motion of the feline antebrachium may increase relative motion between the radius and ulna fracture sites, increasing interfragmentary strain and the potential for major complications such as non-union. Plating the radius and ulna appears to be more accurate in preserving the wide range of supination and pronation of the feline antebrachium. In addition, Preston et al. [34] have shown that radial and ulnar plating in cats is more stable than radial plating alone. Our study confirms that sexual dimorphism in cats manifests itself in radial length. The radius was longer in males by roughly 1 cm. This led us to use a longer plate for radial osteosynthesis in males. The interpretation of the values of aLDRA is more difficult, as the difference that we observed between males and females was on the edge of statistical significance (significant in the whole initial test, but insignificant in the post hoc analysis). Regardless of its true nature, the difference is so miniscule (aLDRA by 1–2° bigger in females) that it may be dismissed as completely insignificant from a clinical standpoint. Far more meaningful is the observation that both radial length and aLDRA show very low diversity among cats, especially when stratified by sex. This is a very promising conclusion for the future widespread use of pre-contoured plates in feline orthopedic surgery. When interpreting the results of this part of our study, it is important to consider its limitations. The desexing status of all cats in our study was unknown, but the majority of cats presented to the veterinary practices were neutered. Generally, male cats in Poland are neutered at a younger age compared to females. Prepubertal gonadectomy has been shown to delay radial physeal closure in domestic cats. This must be taken into account when interpreting and applying qualitative bone measurements. Similar observations were made by Choi et al. [18]. Moreover, all cat cadavers used in the study were of only two closely related breeds (DSH or DLH). This may partially account for a little variability in radius length and aLDRA. Inclusion of a wider range of breeds would undoubtedly increase the variety of anatomical circumstances in which the pre-contoured plate for MIPO surgery could be tested. On the other hand, DSH and DLH are the most common cat breeds in Poland, so the studied cohort is representative of the majority of clinical cases we may encounter in our daily practice.

## 5. Conclusions

The results of this study suggest that the use of the MIPO technique and reduction in comminuted diaphyseal radial fractures under the pre-contoured plate on the 3D-printed bone models of random male and female DSH may be useful in the treatment of complex 22-C2 and 22-C3 fractures in domestic cats. There was no need to pre-contour the plate for each individual case.

## Figures and Tables

**Figure 1 animals-14-01381-f001:**
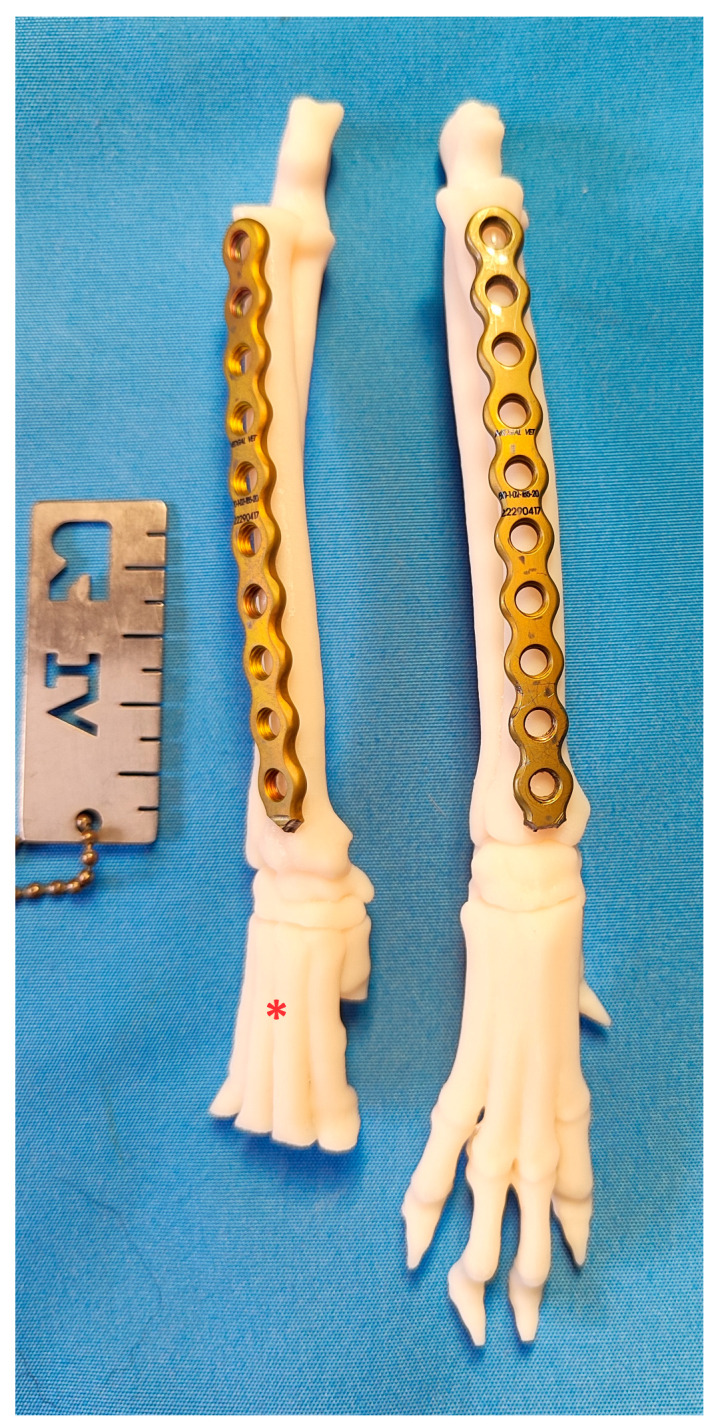
3D printed models of feline (asterisk) and canine antebrachium. The feline radius has a more pronounced lateral torsion than the canine, which is important for radial osteosynthesis in the MIPO.

**Figure 2 animals-14-01381-f002:**
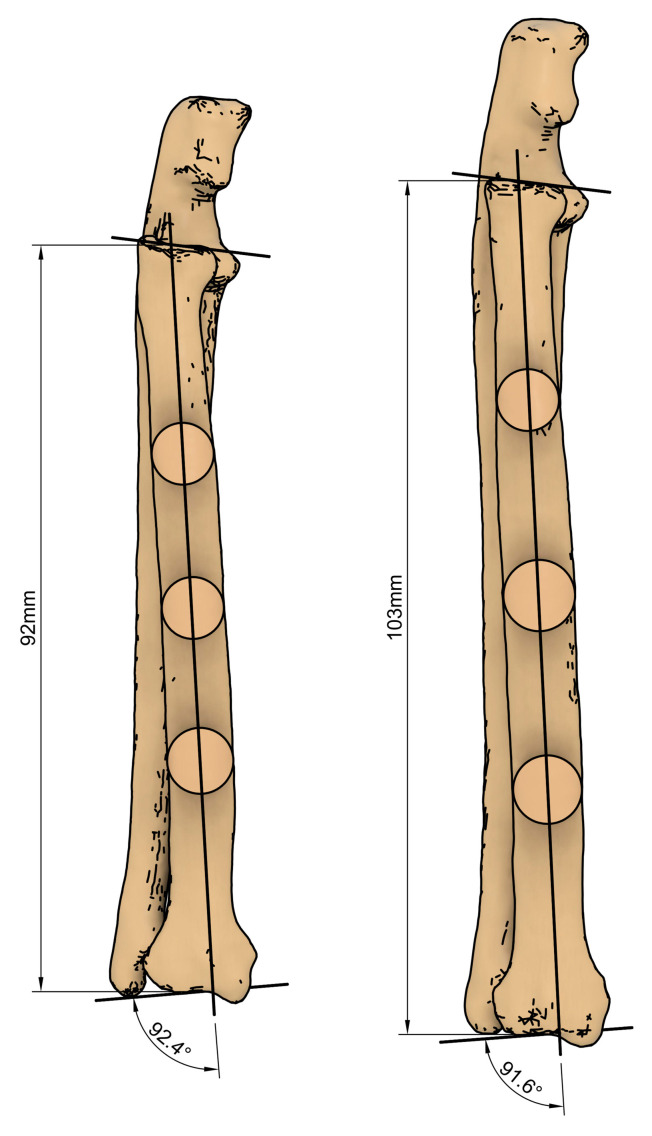
Measurement of the radial length and the anatomical lateral distal radial angle (aLDRA) on the 3D-printed feline antebrachial bone models: 92.4° in the female and 91.6° in the male.

**Figure 3 animals-14-01381-f003:**
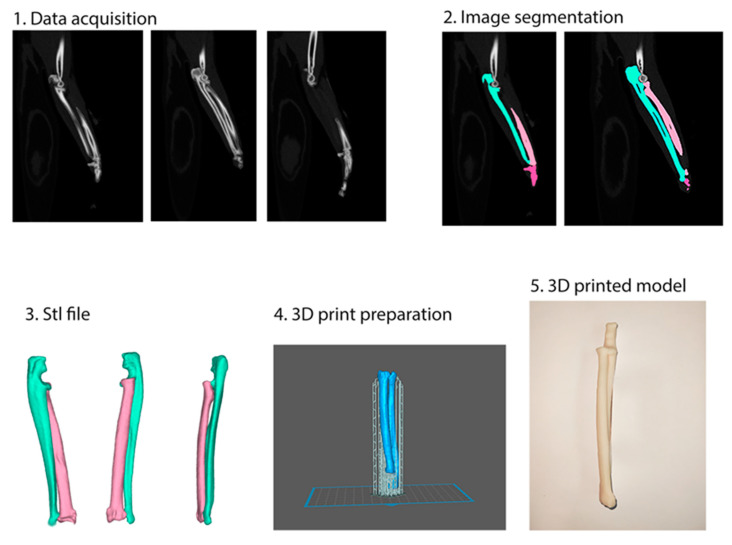
Workflow showing the creation of a 3D-printed bone model of feline antebrachium. 1. Data acquisition from computed tomography. 2. Segmentation of selected parts of the body on the basis of Hounsfield scale bone values. 3. Volume segmentation and exporting to obtain a stereolithographic (STL) file. 4. Preparation for 3D printing: setting up supports, selecting printing material, generating machine codes. 5. Model after the removal of supports and following mechanical cleaning of the surface from support marks.

**Figure 4 animals-14-01381-f004:**
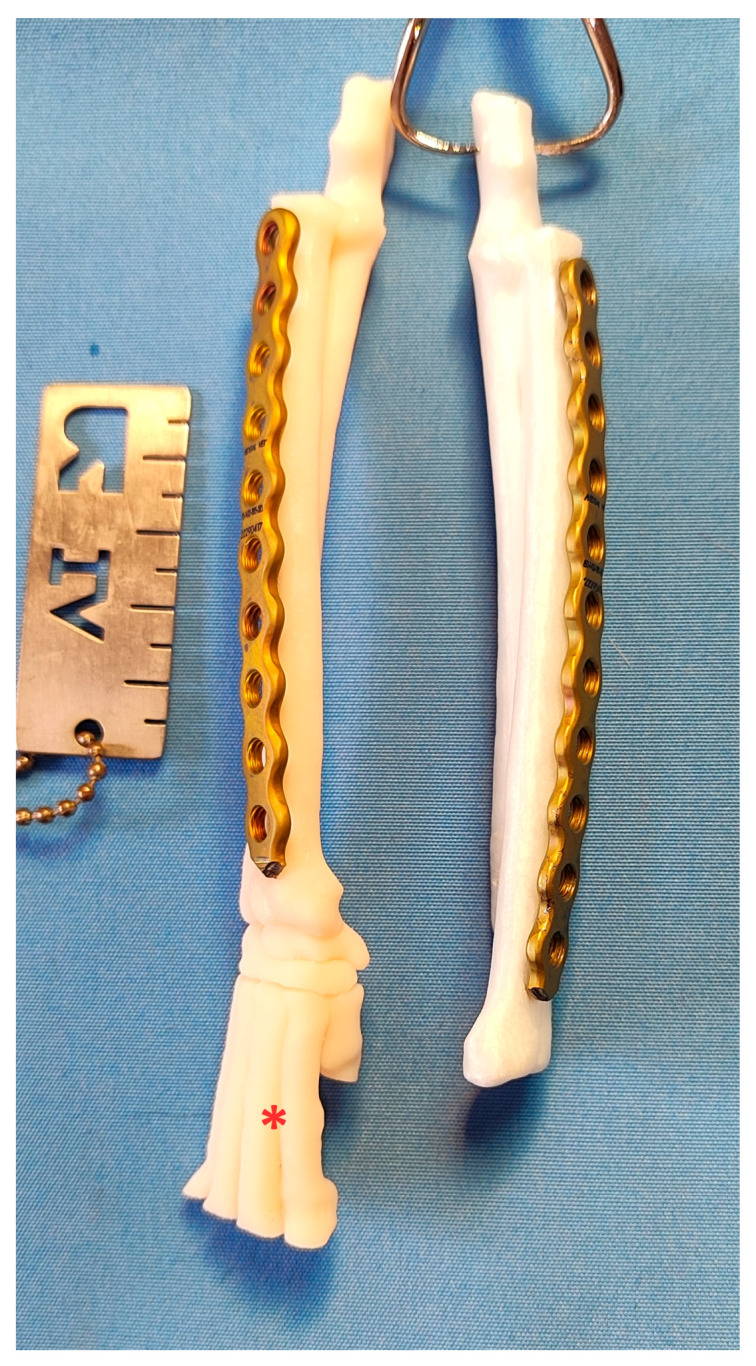
3D-printed models of the left antebrachium of female (asterisk) and male adult domestic shorthair cats. The antebrachial bones of the male cats are longer.

**Figure 5 animals-14-01381-f005:**
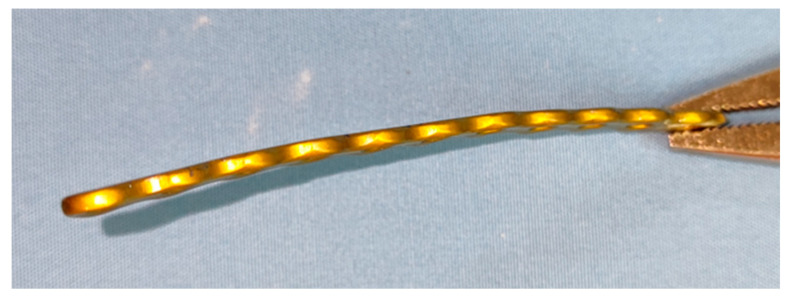
Pre-contoured 90 mm-long 2.4/2.7 locking plate.

**Figure 6 animals-14-01381-f006:**
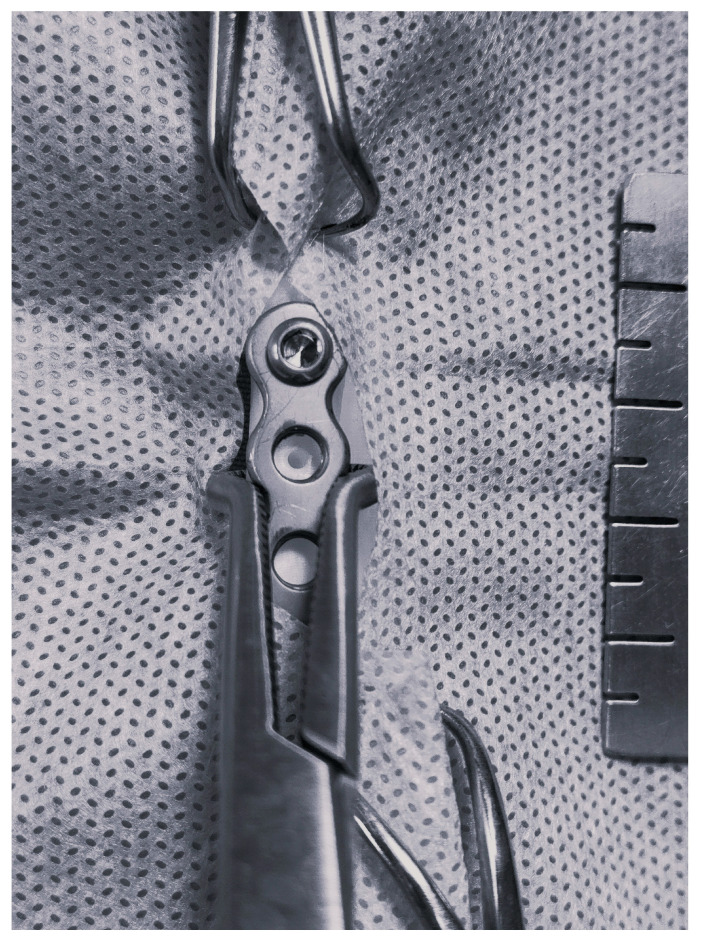
Fixation of the proximal part of the plate to the left radius in a female domestic longhaired cat.

**Figure 7 animals-14-01381-f007:**
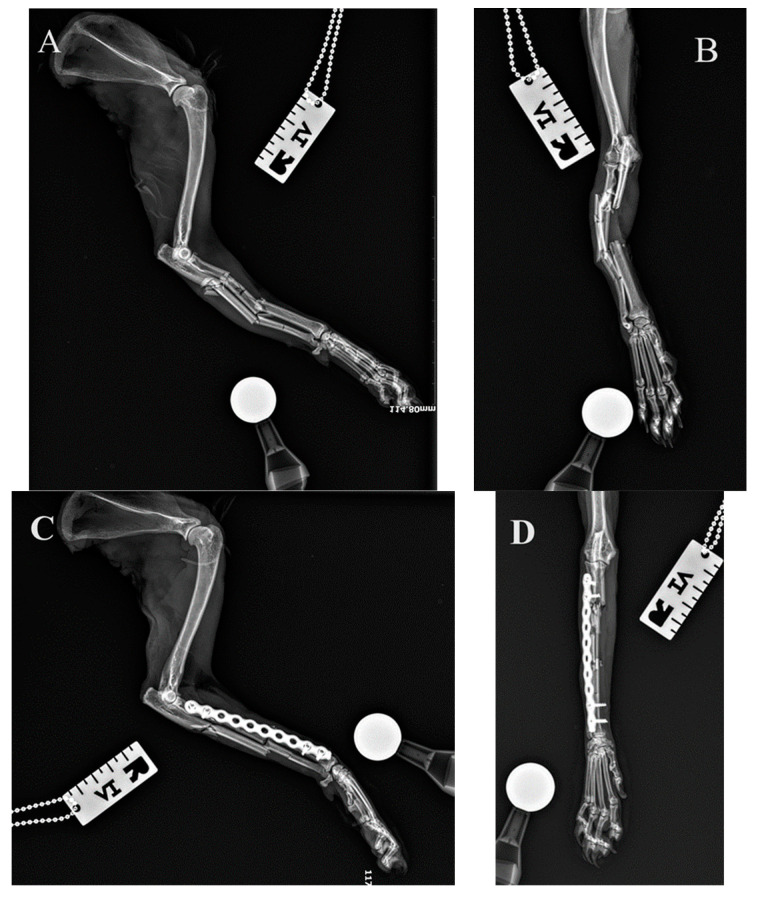
Initial lateral and cranio-caudal radiographs of the right forelimb of a female domestic longhaired cat with a comminuted antebrachial fracture (**A**,**B**). Radiographs after minimally invasive plate osteosynthesis (MIPO) of the radius (**C**,**D**). Correct alignment, adjacency, and apparatus were obtained.

**Figure 8 animals-14-01381-f008:**
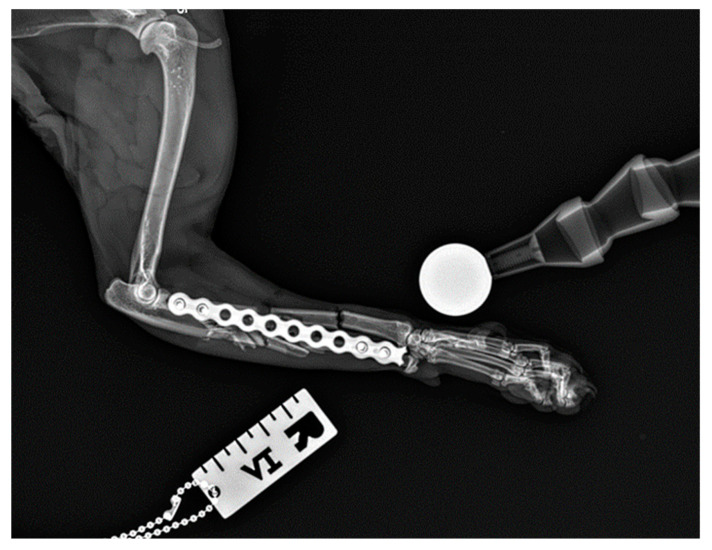
Postoperative lateral radiograph of the right antebrachium of a female domestic shorthaired cat. The plate was fixed by mistake to the proximal radius and distal ulna. The case was immediately revised using minimally invasive plate osteosynthesis approaches.

**Figure 9 animals-14-01381-f009:**
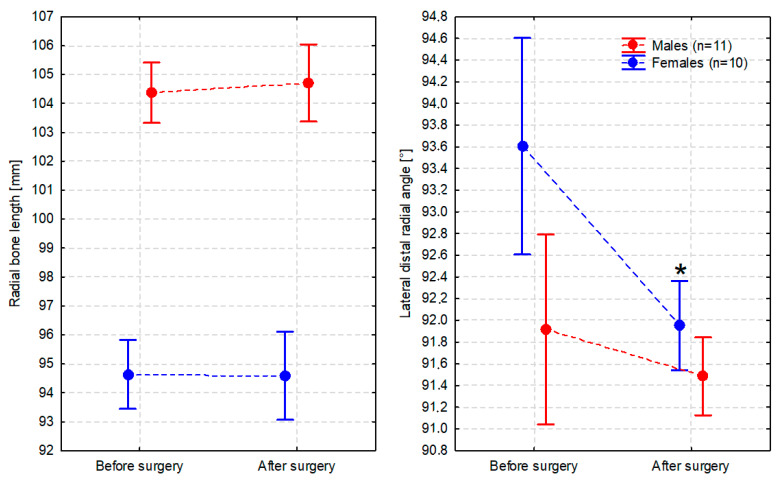
Values of the radial bone length and lateral distal radial angle (averaged for the left and right limbs) before and after surgery in males and females presented as the arithmetic mean (dot) and 95% confidence interval (whiskers). Asterisk indicates statistically significant change (α = 0.05).

**Table 1 animals-14-01381-t001:** Sexual diversity (dimorphism) of radial bone length and lateral distal radial angle (aLDRA) in cats based on measurements taken before the surgery.

Radial Bone Characteristics	Males (*n* = 11)		Females (*n* = 10)		Mean Difference between Males and Females (CI 95%) ^a^	*p*-Value
	Mean ± SD (Range)	CV (CI 95%)	Mean ± SD (Range)	CV (CI 95%)		
Length of left radial bone [mm]	104.4 ± 1.3 (103.0–107.5)	1.3% (0.7%, 1.8%)	94.4 ± 2.1 (92.0–98.0)	2.3% (1.2%, 3.3%)	10.0 ± 1.8 (8.4, 11.7)	<0.001
Length of right radial bone [mm]	104.4 ± 0.9 (103.3–106.2)	0.9% (0.5%, 1.3%)	94.2 ± 2.1 (92.0–97.8)	2.2% (1.2%, 3.3%)	10.2 ± 1.6 (8.7, 11.7)	<0.001
aLDRA left [°]	91.7 ± 1.5 (89.8–94.2)	1.6% (0.9%, 2.3%)	93.4 ± 1.4 (91.3–94.9)	1.5% (0.8%, 2.1%)	-	0.073
aLDRA right [°]	92.1 ± 1.4 (90.5–94.4)	1.5% (0.8%, 2.2%)	93.4 ± 1.3 (91.3–95.0)	1.4% (0.8%, 2.1%)	-	0.225

SD—standard deviation, CV—coefficient of variation; CI 95%—95% confidence interval; ^a^ presented only if statistically significant.

## Data Availability

The data presented in this study are available in article.

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
