# Peer review of "Minimally Invasive Plate Osteosynthesis (MIPO) of Comminuted Radial Fractures Using a Locking Plate Contoured on a 3D-Printed Model of the Feline Antebrachium: A Cadaveric Study"

_animals, 2024, doi:10.3390/ani14091381_

Round 1
Reviewer 1 Report
Comments and Suggestions for Authors
The article was well elaborated, representing a relevant and updated subject, filling a gap in veterinary orthopedics. Nevertheless, the authors should:
- Increase/justify the number of animals used to create the model. They should consider increasing the number of animals to ensure better reproducibility of the results obtained.
- Define what would constitute a good fracture.
- Present Figure 1 adequately in the methodology section.
- Add a reference ruler to all figures.
- Figure 6 does not meet the quality of the work.
- For Figure 7, they should assemble a board to ensure better understanding of the work.
- Specify which normality test was applied.
- Review the conclusions in accordance with the objectives of the stud

Author Response
Dear Reviewer Many thanks for very valuable remarks. Below the modifications of the manuscript. Please see the attachment
Note Increase/justify the number of animals used to create the model. They should consider increasing the number of animals to ensure better reproducibility of the results obtained.
answer: In the Statistical analysis section we added paragraph: “The required number of pairs of forelimbs (from 1 cat) to be used in the study was calculated so that it was possible to detect the minimum difference between two unpaired groups (sex, breed) as well as paired groups (left vs. right forelimb, before vs. after surgery) in the radial bone length by 5 mm (assuming SD of 3 mm in each group) and in the aLDRA by 5° (assuming SD of 3° in each group) with the statistical power of at least 90% (probability of type II error [β] of at most 10%). The minimum size of each group was 9 pairs of forelimbs.”
Note Define what would constitute a good fracture.
answer: A good fracture to treat is fresh, simple and easy to reduce. The soft tissue envelope is not severely damaged and the patient is calm.
Note Present Figure 1 adequately in the methodology section.
answer: : done
Note Add a reference ruler to all figures.
answer: done
Note Figure 6 does not meet the quality of the work.
answer: changed
Note For Figure 7, they should assemble a board to ensure better understanding of the work.
answer: We ask the editor
Note Specify which normality test was applied.
answer: The normality assumption was verified with the Shapiro-Wilk W test. We added this information to the main text
Note Review the conclusions in accordance with the objectives of the study
answer: We rewrite the Conclusions section: “The results of this study suggest that the use of the MIPO technique and reduction of comminuted diaphyseal radial fractures under the pre-contoured plate on the 3D-printed bone models of random male and female DSH may be useful in the treatment of complex 22-C2 and 22-C3 fractures in domestic cats. It was not necessary to pre-contour the plate for each individual case’
Reviewer 2 Report
Comments and Suggestions for Authors
Dear Authors,
This is a very well written paper with a very interesting topic. I only have a few specific comments:
1. Line 16 strike "the " in front of human
Line 22 I would rewrite this to : In our study, we presented results of comminuted feline radial fracture repair using MIPO with locking bone plates as reduction tool.
Line 27 Please change to: Fractures of the radius and ulna are best repaired using internal fixation and stabilization in accordance to AO principles. (please strike after radius: in this species should be treated surgically).
Comments on the Quality of English Language
Hi,
The English has some minor changes that are needed, but editing appears to be minor.
Author Response
Dear Reviewer Thank you very much for your reviews. We have made numerous language corrections
Note Line 16 strike "the " in front of human
answer: done
Note Line 22 I would rewrite this to : In our study, we presented results of comminuted feline radial fracture repair using MIPO with locking bone plates as reduction tool.
answer: changed
Note Line 27 Please change to: Fractures of the radius and ulna are best repaired using internal fixation and stabilization in accordance to AO principles. (please strike after radius: in this species should be treated surgically).
answer: done
Reviewer 3 Report
Comments and Suggestions for Authors
Thanks to the authors for submitting this paper on the use of contoured plates on 3d printed models for MIPO fixation on antebrachial fractures on cats.
There are many things that need to be discussed and addressed before accepting this manuscript as publications
· First and foremost, I struggle to see the novelty of this paper. Yes there are few publications on MIPO on cats (Guiot et 2011, Hudson et al.2009; Bedizci 2020,ecc), but you don’t highlight enough what your paper is bringing to the current literature on feline MIPO. Especially, you started your introduction telling the antebrachial fracture in feline are quite rare, so…? Would have been another bone segment (i.e femur) a better choice?
· So, the novelty of this paper should be the use of 3d printed bones for plate contouring and the use of precontoured plate as reduction tool. However, a severe pitfall in my perspective is that you don’t’ have a control group. That in this case, would be using implants that are contoured directly. And ideally, your tested group (pre-contured implants) should be the one having less surgical time and better reduction. But this Is missing and it weakens a lot your study
· As concerns alignment, you correctly measured the frontal alignment via aLDRA, but there is no mention on torsion correction, which is the biggest challenge when performing MIPO, especially for femoral fractures, but can be also a concern for the radius. This is particularly true in cats, where you correctly stated that feline have increased radial torsion compared to dogs. Why did you not evaluate the radial torsion?
· English has to be improved
· In lines 81-83 you talk about “clinical observation”: this is not objective and supported by literature, therefore has to be removed
· Lines 168-170: this is not clear: how can you applied traction to the fracture if you have already inserted 2 locking screws, one per each bone fragment?
· There are some general terms that need to be changed throughout the manuscript, as they are not routinely used for MIPO description. For example:
o Submuscolarly tunnel: epiperiosteal tunnel
o Line 160: distal approach to what? You need to be precise with terminology
Comments on the Quality of English Languageneed robust improvment
Author Response
Dear Reviewer
At firs many thanks for the critical review and valuable suggestions. They will definitely improve our paper. Below the modifications of the manuscript. Please see the attachment. We have made numerous language corrections
Note
First and foremost, I struggle to see the novelty of this paper. Yes there are few publications on MIPO on cats (Guiot et 2011, Hudson et al.2009; Bedizci 2020,ecc), but you don’t highlight enough what your paper is bringing to the current literature on feline MIPO. Especially, you started your introduction telling the antebrachial fracture in feline are quite rare, so…? Would have been another bone segment (i.e femur) a better choice?
answer: In fact, antebrachial fractures are less common in cats than, for example, femoral fractures. However, in our clinical practice we find that antebrachial fractures in cats are underestimated. Many clinicians who treat such fractures in dogs treat cats in a similar way without considering the differences between the two species. Cats have unique structural and functional characteristics of the forelimb that may influence clinical outcomes after repair of antebrachial fractures, e.g. the feline radius has a more pronounced lateral torsion than the canine radius, which is important for osteosynthesis of radial fractures. In the Introduction section of our study, we described such differences between cats and dogs to better understand the difficulties in treating antebrachial fractures in feline patients. In our opinion, the treatment of fractures of other long bones in dogs and cats is similar, but the feline antebrachium deserves more attention, and therefore our study focused on testing facilitators to improve the quality of osteosynthesis in prospective clinical cases. For more clarity we rewrite the aim of our study:
“In our clinical practice we have found that the antebrachial bones of domestic cats are very similar in length and shape between individuals. We believe that the bone plate used as a reduction tool does not need to be pre-contoured for each individual. The aim of our study was to present the results of the reduction of 42 comminuted feline radial fractures in the MIPO using locking bone plates as a reduction tool. The plates were pre-contoured on 3D printed bone models of the antebrachium of a random adult male and female DSH.
Note So, the novelty of this paper should be the use of 3d printed bones for plate contouring and the use of precontoured plate as reduction tool. However, a severe pitfall in my perspective is that you don’t’ have a control group. That in this case, would be using implants that are contoured directly. And ideally, your tested group (precontoured implants) should be the one having less surgical time and better reduction. But this Is missing and it weakens a lot your study
answer: Our study was designed to test the hypothesis that adequate reduction of comminuted feline radial fractures in the MIPO mode using locking bone plates as the reduction tool is possible after pre-contouring the implants on 3D-printed bone models of the antebrachium of random adult male and female domestic cats. Our clinical observations suggest that there are no differences in radius and ulna shape between adult domestic shorthair (DSH) and longhair (DLH) cats, so we do not create a control group. We think that your suggestion is appropriate and we consider it to be an additional limitation of our study. We have added a paragraph on this limitation in the Discussion section:
“. We did not create a control group of cats with pre-contoured plates for each radius. Comparing two groups of animals, one treated with plates pre-contoured on 3D-printed models and one treated with plates pre-contoured for each individual, may provide valuable results and may be the subject of further studies”
Note As concerns alignment, you correctly measured the frontal alignment via aLDRA, but there is no mention on torsion correction, which is the biggest challenge when performing MIPO, especially for femoral fractures, but can be also a concern for the radius. This is particularly true in cats, where you correctly stated that feline have increased radial torsion compared to dogs. Why did you not evaluate the radial torsion?
answer: As you rightly said, rotational malalignment is one of the main problems in performing MIPO. On the pre- and postoperative radiographs, we only measured the length of the radius and the aLDRA. The axial and rotational alignment of the bone fragments was assessed during flexion and extension of the elbow and carpal joints. In addition, alignment (axial and rotational alignment of the joints above and below the fracture) was assessed on postoperative radiographs. This is standard clinical practice. Modern imaging techniques, such as computed tomography, should be used for more adequate postoperative assessment of radial torsion. We did not use computed tomography for postoperative assessment and this could be seen as a potential limitation of our study. We have added a paragraph on this limitation in the Discussion section:
“As torsion of bone fragments is one of the main difficulties in MIPO, more advanced imaging techniques such as computed tomography may be used for more appropriate postoperative assessment of bone alignment. In our study, we did not use computed tomography for postoperative assessment and this could be considered a potential limitation of this study”
- Note English has to be improved
Note In lines 81-83 you talk about “clinical observation”: this is not objective and supported by literature, therefore has to be removed
answer: done
Note Lines 168-170: this is not clear: how can you applied traction to the fracture if you have already inserted 2 locking screws, one per each bone fragment?
answer: The plate was first applied to the proximal end of the radius. After manual traction of the antebrachium and manipulation of the paw, the radial fracture was reduced under the plate and the distal end of the plate was screwed to the distal fragment of the radius. The absence of post-traumatic changes such as swelling, haematoma and skeletal muscle contracture was described as a limitation of our study, as reduction of fractures in cadavers without these changes was easy. Manual traction and manipulation of the bone fragments was sufficient for good reduction of all fractures. Line 366-368
Note There are some general terms that need to be changed throughout the manuscript, as they are not routinely used for MIPO description. For example:
- Submuscolarly tunnel: epiperiosteal tunnel
answer: done
- Line 160: distal approach to what? You need to be precise with terminology
answer: We added sentence “to the radius”
Reviewer 4 Report
Comments and Suggestions for Authors
Dear authors,
It is a very interesting study, where it is shown that having a plate that can be already contour to the bone surface and length, could help in reducing surgical time and other implant related problems such as too short or too long bone plates.
I have few comments regarding more the discussion rather than the study itself:
-In cats, does the olecranon and ulna, do they do not need to be fixated as well? your research only shows radial fracture reduction and stabilization with a plate, but what happens with the ulna? is it not a prognosis indicator as well?
-What are the consequences if the ulna and radius are fixed together as you mention on lines 216-217. During the healing process, if the ulna is left with a non reduced fracture, does the excessive bone callus affects the radial healing? and the limb function?
-Line 242 is written "an fracture" it should read "a fracture"
-What is it that should be done if the fracture not only affects the diaphysis but also the physis or epyphisis? Does your described method could still me applied?
-Finally, this study is consider for cats with similar body height, with a very specific fracture type. You explain the low prevalence of this types of fractures, but as body sizes can vary, how applicable would it be, in reality, as having a specific plate, with a fix length, on a general population of cat patients?
Comments on the Quality of English LanguageNo specific comments.
Author Response
Dear Reviewer Many thanks for very valuable remarks.
-In cats, does the olecranon and ulna, do they do not need to be fixated as well? your research only shows radial fracture reduction and stabilization with a plate, but what happens with the ulna? is it not a prognosis indicator as well?
answer: Fractures of the radius and ulna in cats should be treated surgically to reduce the risk of malunion or nonunion. In our study, the ulna was not reduced and stabilised to test the usefulness of a contoured radial plate as a reduction tool, but in clinical practice we treat such fractures by plating the radius and ulna.
-What are the consequences if the ulna and radius are fixed together as you mention on lines 216-217. During the healing process, if the ulna is left with a non-reduced fracture, does the excessive bone callus affects the radial healing? and the limb function?
answer: Among domestic animals, only cats are able to actively perform significant supination movements. Cats need to maintain their antebrachial range of motion to perform daily functions. Fixation of the radius and ulna together will prevent the ability to perform supination movements. Similarly, callus formation between the radius and ulna can also limit supination and pronation of the antebrachium.
-Line 242 is written "an fracture" it should read "a fracture"
answer: done
-What is it that should be done if the fracture not only affects the diaphysis but also the physis or epyphisis? Does your described method could still me applied?
answer: To stabilise such fractures, we use T-plates instead of straight plates. In our clinical practice, we have also found that a precontoured T-plate on the bone model is a useful tool for reduction and stabilisation of physeal fractures.
-Finally, this study is consider for cats with similar body height, with a very specific fracture type. You explain the low prevalence of this types of fractures, but as body sizes can vary, how applicable would it be, in reality, as having a specific plate, with a fix length, on a general population of cat patients?
answer: As you may have noticed, we completed the group of domestic shorthair (DSH) and longhair (DLH) cats with similar body weight and length of the radius. The DSH and DLH are the most common breeds of cats in Poland, so the cohort studied is representative of the majority of clinical cases we may encounter in our daily practice.
Round 2
Reviewer 3 Report
Comments and Suggestions for Authors
Thank to the authors for the revision. I think that manuscript has improved I terms of quality and preparation. I still struggle to see a great novelty for this study, but because 3d printing and MIPO are hot topic in veterinary medicine, I guess there is still room for this paper to be published
I would remove from the paper any personal considerations, only objective data needs to be mentioned. For example, at the line 89 you stated: " in our clinical practice" - please remove it and soon on.
Comments on the Quality of English LanguageEnglish has improved but needs further improvement. An editing service or English mother tongue co-authors is reccomended
Author Response
Dear Reviewer
Once again many thanks for very valuable remarks. Below the modifications of the manuscript. Please see the attachment
- we removed line 84-88
"Our clinical observations suggest that there are no differences in radius and ulna shape between mature shorthair (DSH) and longhair (DLH) domestic cats. We have, however, observed that males have longer antebrachial bones. Choi et al. [18] and Boonsri et al. [21] made similar observations."
We add
"In domestic cats, the radius and ulna are similar in length and shape between individuals, with the exception that the radius is longer in males [18, 21]. Therefore, we believe that the bone plate used as a reduction tool does not need to be pre-contoured for each individual. The aim of our study was to present the results of the reduction of 42 comminuted feline radial fractures in the MIPO using locking bone plates as a reduction tool. The plates were pre-contoured on 3D-printed bone models of the antebrachium of a random adult male and female DSH"
- we removed line 321
"In our clinical practice, we always try to use three bicortical locking screws per fragment"
We rephrase line 322-325
"Palierne et al. [30] compared plates fixed with two and three locking screws per fragment and found that the bending stiffness of the 2-screw constructs was lower than that of the 3-screw constructs. Therefore, it is better to always try to use three bicortical locking screws per fragment."
- We removed line 403
"In our clinical practice, we see"
We add
"Such complex fractures are most commonly seen in cats that have suffered serious trauma, such as falls or road traffic accidents. In such patients, the MIPO technique is always favorable for the reduction and stabilisation of these fractures, as MIPO is less time consuming and less traumatic than open reduction and internal fixation (ORIF) [13, 14, 15]. "
- we removed line 394
"In in our authors’ clinical practice experience,"
We rephrase line 393-397
"In this study, ulna was not reduced and stabilized in order to test the usefulness of a contoured radial plate as a reduction tool however, ulna fractures in cats should be stabilized. In most cases, the ulnar plates may be a better choice than intramedullary pins."
We removed line 411
"We that double plating is more accurate in preserving the wide range of supination and pronation in cats."
- We rephrase line 429
"This is a very promising conclusion in terms of a future wide use of pre-contoured plates in feline orthopedy"
on
"This is a very promising conclusion for the future widespread use of pre-contoured plates in feline orthopaedic surgery."
- We rephrase in Statistical analysis Line 208-216
"The required number of pairs of forelimbs (from 1 cat) to be used in the study was calcu-lated so that it was possible to detect the minimum difference between two unpaired groups (sex, breed) as well as paired groups (left vs. right forelimb, before vs. after surgery) in the radial bone length by 5 mm (assuming SD of 3 mm in each group) and in the aL-DRA by 5° (assuming SD of 3° in each group) with the statistical power of at least 90% (probability of type II error [β] of at most 10%). The minimum size of each group was 9 pairs of forelimbs. The statistical analysis was performed in the TIBCO Statistica 13.3 (TIBCO Software Inc., Palo Alto, CA, USA)."
on
"The minimum size of each group analyzed in the study was 9 pairs of forelimbs. This group size ensured ≥90% statistical power of comparisons of the radial bone length and the aLDRA between unpaired groups (i.e. males vs. females, DSH vs. DLH) as well as be-tween paired groups (left vs. right forelimb, measurements before vs. after surgery). The minimum difference that could be detected at this group size was a difference in the mean radial bone length between groups equal to 5 mm (assuming SD in each compared group equal to 3 mm) and a difference in the mean aLDRA between groups equal to 5° (assuming SD in each compared group equal to 3°). The statistical analysis was performed in the TIBCO Statistica 13.3 (TIBCO Software Inc., Palo Alto, CA, USA)."
